# Annotating Sexism in Online Political Discourse: Assessing the performance and confidence of LLMs

## Abstract

Large Language Models (LLMs) have recently gained popularity for text analysis within the social sciences due to their versatility and context-aware capabilities. The use of prompt-based learning of LLMs has especially increased its application in classification tasks and text annotation of sensitive topics like sexism. While studies have used them for capturing online sexism, not much has been known of their capabilities across lesser-known discourses like that of political discourse, and how the models distinguish between partisan bias to gender bias. In this research, our main contributions could be listed as: i) comparing different LLMs through prompt engineering in their capability of detecting sexism in political discourse; and ii) proposing a new algorithm for capturing the confidence of the LLM predictions in classification tasks. Experimental results demonstrate a clear indication of trigger events that provoke online sexism, and yet no clear advantage of using LLMs while predicting sexism. Surprisingly, the results do not improve with more instructive prompts, but our algorithm proves to be effective in capturing the confidence of each model on their predicted labels.

**Content warning:** This document studies contents that may be offensive or upsetting. It will have illustrative examples of sexist languages online.

## 1 Introduction

> "If Hillary Clinton can't satisfy her husband what makes her think she can satisfy America?"          - Donald Trump (Twitter, 2015)

We ask the readers if they consider the above text to be sexist or not. There is no slur or backhanded compliment, and it does not disqualify Hillary Clinton based on her being a woman. Yet Clinton was clearly sexualised in this comment. The above example provides a perfect illustration of how difficult it can be to determine whether a piece of text is sexist or not. Often, this task involves relying on information not present in that text, such as understanding who the speaker is and in what context the speaker is making that statement. In politics, sexist discourse can often appear alongside criticisms to a given party or candidate. In fact, Ozer (2023) claim that partisan polarization [or differences] has shown to exacerbate gender-based stereotypes and biases. In fact, Lupu et al. (2023); Kalyanam et al. (2016) find that offline trigger events, such as protests, elections and news, are often followed by increases in online hate speech that bear seemingly little connection to the underlying event. Especially, hate speech direct at targeted social groups are shown to spike in X (formerly Twitter) after such 'trigger' events for a period immediately following the said event (Burnap & Williams, 2016). This affect is also extended (and in certain case, elevated) to female politicians after any such trigger events. An article from BBC (BBC, 2022) revealed that some of the female politicians came off the platform because of the same reason, which eventually impacted their public engagement. Due to the widespread impact of social media in politics (Reveilhac & Morselli, 2023) and the surge in online hate speech (including sexism), it is important to study more of such propagation in political science research. Grimmer & Stewart (2013) states that recognizing the language is central to the study of political texts, and automated methods (models) used by researchers to handle massive political texts and make inferences are usually *incorrect* as the models fare well when the texts fit the assumption of the models, hence require careful validation.

Given the huge resources of online data in the Internet, such as news articles and social media, we can study sexist discourse at scale. There are several existing datasets, from sexism-specific datasets (such as Samory et al. (2021); Melville et al. (2019); Jha & Mamidi (2017); Kirk et al. (2023), *inter alia*) to general political discourse (such as Lindgren & Åkerlund (2022); Reddit (2022); Battaglia & Caliendo (2022); CrowdFlower (2016), *inter alia*), but to our knowledge, not any research on the sexism in online political discourse. Ziems et al. (2024); Burnham (2024) has stated that political ideology [like quantifying real and perceived political differences] as one of the complex social phenomenon. Political statements directed at women politicians are mistaken as sexist because the emotion/tone in political discourse usually tend to be negative (given the tone and phrasing of the text §C.1) and if this negative emotion is directed at women, it is mistakenly classified as sexist. This is because theoretically and linguistically, political discourse has emotional conflict such that it is often difficult to differentiate between the political differences and sexism. Furthermore, now we also have powerful Natural Language Processing (NLP) tools which are utilized for identifying online sexism, using unsupervised approaches like topic modelling (such as Melville et al. (2019)), and supervised approaches with several hate speech classifiers (such as Samory et al. (2021); Jha & Mamidi (2017), *inter alia*) that uses word embeddings, which are specifically designed to capture sexism. However, as can be seen from the example tweet above, the text to be classified often does not offer much content or context which a topic model or embedding-based approach could use. Furthermore, most research rely on identity terms and lexical dependencies which eventually result in false positives (or *false alarm*) and severe unintended biases (Dutta et al., 2024). Yet, LLMs has offered unprecedented opportunities to explore the nature of intelligence, language and thought due to their remarkable performance on various cognitive tasks (Niu et al., 2024).

As LLMs are being adopted globally by users with diverse backgrounds, it is expected that they are designed to reflect their values and preferences (Kirk et al., 2024; Dong et al., 2024). Likewise, studies like Ziems et al. (2024); Linegar et al. (2023) recommend political science and computational social science researchers to consider using LLMs as foundation of annotation tasks, given its strength at generation tasks and producing superior annotations than human gold annotators for over 38% of the times in the tasks they evaluate. Linegar et al. (2023) situate that LLMs can replace manual annotation efforts (particularly in processing political content) because of their increased ability of information extraction as compared to other NLP algorithms. Though not fully replaceable, LLMs can potentially reduce the cost and time required for annotation (Thapa et al., 2023). In general, LLMs have shown to exceed human performance of reliably classifying texts in some domains, without the need for supervision (Ziems et al., 2024; Burnham, 2024), which make them an efficient candidate as an annotation tool. This has led to its widespread adoption among political and social science researchers with its use through prompt engineering using zero-shot and few-shot learning (Burnham et al., 2024). Prompting as annotators has therefore shown positive results in multiple automated tasks (Brown et al., 2020). Tan et al. (2024) claim LLMs as an excellent alternative to crowd-sourced annotators and can significantly mitigate the challenges encountered with traditional annotation methods.

Argyle et al. (2023) demonstrates how generated responses from LLMs could be indistinguishable from parallel human texts through human evaluators, arguing that the problematic social biases such as sexism can be treated as uniform properties of the model. However, since these LLMs are trained on massive corpus, these biases present in their training datasets can create harm with biased representations (Webster et al., 2020; Nangia et al., 2020). The quality of the model's annotation, based on their prediction capability, can directly impact the reliability of these models. This leaves us to question how much *should we* trust LLMs on their usage as human replacements in the same social biases. While studies have already started using LLMs for annotation task, there has been limited studies (such as Mohta et al. (2023)) which question the trustworthiness of the LLMs, and mostly focusing on images or multi-modal inputs, or on natural language generation (such as Kuhn et al. (2023)). We particularly aim to question the LLMs' trustworthiness exclusively in NLP tasks through the model's confidence and propensity to incorrect predictions in a domain-specific setting. Current NLP research focuses on the model's predictive confidence (or uncertainty) and calibration to access its predictive performance (Xu et al., 2024). We further assess its confidence by proposing a simpler method called relative entropy in the classification task, based on the model outputs through different iterations.

Overall, this research investigates the challenge LLMs face in capturing different forms of sexism in online political discourse, where one's political position may often be intertwined with gender

bias. We therefore question how capable LLMs are in capturing sexism in online political discourse, with different levels of instructions provided to them through prompting? **(RQ1)** Can we device an algorithm to improve evaluation of the confidence of LLMs in their predictions? **(RQ2)** And are the LLM predictions trustworthy? **(RQ3)** We propose a new algorithm that can efficiently test the confidence of predictions by any LLMs. Recently, researchers have questioned the usefulness of using LLMs as an alternative to text annotations in political science research. In our research, we provide concrete evidence to the research community in the effectiveness of LLMs in performing classification tasks of sensitive topics like sexism, when presented with domain-specific discourse.

## 2 EXPERIMENTAL SETUP

### 2.1 DATASET

#### 2.1.1 DATA OVERVIEW

We collected data from X (formerly Twitter), using their official application programming interface (API) through academic access, for the year of 2022 and based in the United Kingdom (UK). The year of 2022 was chosen for our study because it saw a lot of political and economic developments in the UK, with three changing Prime Ministers within a short span of time, the death of Queen Elizabeth II and a deepening economic crisis (Middleton, 2023). The intention was to consider the time which would have more political and non-political trigger events which we can analyze for the propagation of multiple types of sexism at different points, and check their trend along the way. Initially, we identified 38 female Ministers of Parliament (MPs) from the United Kingdom, based on their political positions and online activity on X. This was done by monitoring their profiles and public engagement online, hence ensuring that the selected MPs actively use the platform to connect with the public. The number was later brought down to 3 female politicians, namely *Angela Rayner, Liz Truss* and *Suella Braverman*, based on the reasons as explained in §2.1.4, and data was collected with using their names and usernames as the keywords, yet excluding posts from the usernames themselves. We selected different time-periods of 2022 based on any known political or controversial event centering around any of the said politicians. For collecting relevant tweets for our study, we only considered the reply tweets[1] since we want to do a computational analysis on detecting sexism based on the opinions, emotions and attitudes of the public centering around the mentioned politicians. The collected tweets did not contain tweets posted by the targeted female MP, as the intention was to analyze the conversations *about* them, but *not* by them. Similarly, it does not contain retweets (i.e., re-posting of the original tweet shared by the original user's followers) as well, since it was seen more beneficial for analyzing the virality and propagation of the original tweet and in the analysis of users posting them, both of which do not add value to the type of conversations we focused on for this research. Additionally, it was also causing duplication of a lot of entries. As a result, we removed them from our study.

#### 2.1.2 DATASET PREPARATION

We cleaned the text obtained from X using multiple pre-processing techniques to minimize the noise existing in our data, which accounted for incomplete information that could have resulted in faulty classification; and maximize the understanding of the text by both annotators and models. To minimize the noise, we performed the following steps in respective sequence: (1) dropping empty entries or extra spacing; (2) dropping duplicates; (3) dropping non-English texts; (4) dropping data containing only URLs or emojis (due to the vast number of emojis, one could leave their meaning based on the user's interpretation, hence they can be confusing to the classifiers); (5) remove news articles or posts mentioning the political figures through a set of keywords[2]; (6) expanding contracted texts and changing emojis to text emoticons. Post data cleaning, we sampled entries that contained most number of engagements among the public. As the metrics of engagement, we considered sorting our data in terms of the highest number of 'retweet_count', 'reply_count', 'like_count', 'quote_count' values[3] and considering the entries with highest number of engagements, with respect to the trigger events for our annotation and analysis.

---

[1]The different types of tweets expected from the Twitter (X) platform: `https://shorturl.at/OMR6t`

[2]This is to ensure that we focus only on the user behavior in our data, and not that of any institutions like newspapers or corporations. *e.g., 'BREAKING NEWS', 'HEADLINES:', 'In today's news', etc.*

[3]These units are present in metadata of the original data, which would not be shared publicly in GitHub.

### 2.1.3 DATA ANNOTATION

Post data preparation, the data was annotated by a group of seven experts (three male and four female) who work on gender studies in political science. We conducted the annotations in two phases – (i) one annotation per instance, and (ii) three annotations per instance, where we considered the minority voting scheme[4]. As annotation guidelines, the experts were given a comprehensive document describing the research objectives, consisting of the definitions of political and sexist attitudes we took in our research, similar to what we feed to the model prompts. They were also provided with multiple examples of possible ambiguous examples, and instances where sexism is (in/)distinguishable from political differences, and where they co-exist. We define the two terms as:

(A) *Sexist*: A text is sexist if the speaker shows a prescriptive set of behaviors or qualities that women (and men) are supposed to exhibit in order to conform to traditional gender roles. This could be texts formulating a descriptive set of properties that supposedly differentiates the two genders, portrays women as less competent and less capable than men, and expressed through explicit or implicit comparisons and perpetuating gender-based stereotypes.

(B) *Political*: Texts revolving around discussions of politicians, policies, government actions, ideologies, elections, etc. These texts would aim to engage with societal issues, power dynamics, and decision-making processes within the realm of public affairs. Pertaining to the practice and theory of influencing other people on a civic or individual level, often concerning government or public affairs. A typical political text could have strong language, a harsh tone and slurs; and question the political standing or ideological positions of politicians or public officials.

### 2.1.4 DATASET STATISTICS

| Type | Date | Event |
|---|---|---|
| Sexist | *25th April* | Angela Rayner was the subject of a report in The Mail on Sunday, by Glen Owen, in which it was alleged that she had tried to distract Boris Johnson in the Commons by crossing and uncrossing her legs in a similar manner to Sharon Stone in a scene from the 1992 film Basic Instinct. |
| Political | *6th September, 24th October* | Liz Truss was appointed as Prime Minister by Queen Elizabeth II at Balmoral Castle on 6th September. She was succeeded by Rishi Sunak as leader of the Conservative Party on 24 October. |
| Political | *25th October* | Suella Braverman was reappointed as the Home Secretary by Prime Minister Rishi Sunak upon the formation of the Sunak ministry. Braverman's reappointment was challenged by Labour Party MPs, Liberal Democrats, Scottish National Party MPs and some Conservatives. |

Table 1: This table presents the four trigger events considered in this study, along with their respective dates. The controversies centering around the target female politicians are also mentioned. Though the conversations were mostly centered around the said politicians, mentions of the other politicians we considered were also found in our data.

| Political trigger event | | |
|---|---|---|
| Not sexist | 539 | 95.23% |
| Sexist | 27 | 4.77% |

| Sexist trigger event | | |
|---|---|---|
| Not sexist | 624 | 88.26% |
| Sexist | 83 | 11.74% |

Table 2: Our dataset statistics showing the total number of instances for each label, along with their distribution percentage, at the event of their respective trigger types.

Post data collection, **four** incidents of 2022 were chosen as our *'trigger events'*, based on the sheer volume of conversations collected during those times. Inspired from Kalyanam et al. (2016), we define trigger events as *events relating to the targeted individual(/s) that have stirred up heightened media attention, public scrutiny, social media engagement [high activity] in online platforms with conversations centering around the said individual(/s) and has the potential to trigger sexist comments*. The conversations were collected on the day of the said incident. We attributed the four different trigger events to two trigger types (see Table 1 and 2) by identifying groups of events that produced more concentration of high-activity than other events. While the sexist and political trigger types are targeted at female politicians that can potentially lead to sexism, we also aimed to compare the ability of LLMs in detecting sexism based on the type of trigger events, hence having an impact due to the period of propagation and the trigger event type.

Abusive content online constitutes a minimal percentage of all the posts. Measurement studies from academics and thinktanks indicate that 0.001% to 1% of content on mainstream platforms contains abuse (Vidgen et al., 2019). Among these, sexism forms an even smaller portion, since abuse itself can be of various kinds. In our dataset of *n=1,273* (Table 2), we find much more sexist content than the usual measure for both instances. This leads us to investigate if more conversations around

---

[4]In this annotation scheme, the minority label gets a preference over the majority voting. More information on the annotation task in §A.

targeted female politicians could potentially lead to an increase in online sexism, and therefore we chose to compare between these trigger events in our quantitative analysis (§3.1).

## 2.2 LLMs for Prompt Engineering

### LLMs Used

We selected **five** LLMs for our research, namely Alpaca-7B[5], ChatGPT-3.5-turbo[6], Flan-T5-xl[7], Mistral-7B[8] and Vicuna-7B[9]. Our intention was to test the capabilities of LLMs using the most cost-effective way, i.e., open-source models and without access to graphics processing unit (GPU). Aside from that, using open LLMs promote inspectability and transparency in research, by allowing them to view their built, architecture and specific settings of hyperparamters used - all of which are integral in the performance of larger models and their capabilities of handling complex tasks, such as online sexism. To compare between the models, we also worked with a closed-source model: ChatGPT. This is to ensure a comparison of the performances among the various sources of models.

### Prompt Stability

Responses from LLMs are usually susceptible to the influence of the choice of the prompts (Griffin et al., 2023), and we had seen that initially in our work as well. To ensure that our prompts are robust, we used several prompt settings, with measuring the variation of the performances among several prompt structures. We started off with simple examples that the LLMs had to validate as sexist or not, gradually progressing towards difficult instances (i.e., some selected ambiguous instances where the presence or absence of sexism is difficult to identify). Furthermore, post collection of ground-truth (annotations) in our dataset, to evaluate the prompt effectiveness based on the output quality of the respective LLM, we used four kinds of prompt evaluation metrics[10]: (i) grounding (the authoritative basis of the LLM output, determined by comparing it against some ground truths in a specific domain), (ii) relevance (how relevant the LLM's response is to a given user's query), (iii) efficiency (the speed and computing consumption of the LLM to produce the output.), and (iv) hallucinations (looking at LLM hallucinations with regard to retrieved context). For most of the models, all of these four metrics gave positive outcome within a few trials (except Mistral).

### Prompt Structure and Model Selection

We developed a general template of prompt, which we re-used in all the LLM prompt categories (which are elaborated in §2.2.1 and the instructions are detailed in §D), adjusting according to the required context length for each LLMs, as per their instruction strategies. Due to the limitation in resources, we only worked with the GGUF versions[11](Ggerganov, 2023) of all the LLMs, except for the ChatGPT-3.5 turbo, where we used their API access. We encountered huge differences in the prompting structure, mainly due to the limited context length for some (e.g., Alpaca and Vicuna). This resulted in limited descriptive capacity for the prompts, which in turn affected the understandability of the models when presented with more instructions on the classification task and representative examples. Consequently, Weber & Reichardt (2023) too underscores the need for considering both the nature of the annotation task and the characteristics of the models when designing prompts. We therefore record the context length alongside each LLMs and share the prompts for each models used in our GitHub repository[12].

### 2.2.1 Prompt Categories

We use four different prompt categories to test the understandability of the model. Though the prompts used in each model are uniquely different, they follow a similar template[13]. The prompts are designed in a way that the model would be able to differentiate well between the gender bias

---

[5] https://huggingface.co/TheBloke/claude2-alpaca-7B-GGUF

[6] https://platform.openai.com/docs/models/gpt-3-5-turbo

[7] https://huggingface.co/google/flan-t5-xl

[8] https://huggingface.co/TheBloke/Mistral-7B-v0.1-GGUF

[9] https://huggingface.co/TheBloke/vicuna-7B-v1.5-GGUF

[10] *"How to measure the quality of LLMs, prompts and outputs"* Source: https://shorturl.at/MNKQt

[11] "GGUF is an advanced binary file format for efficient storage and inference with GGML. A model quantized with GGUF will usually have the quantization information in its name, e.g., Q4_0 means that the model is quantized to 4-bit (INT4). In terms of accuracy and model size, they are very similar to GPTQ." Source: https://kaitchup.substack.com/p/gguf-quantization-for-fast-and-memory

[12] All code and data for this work is stored in the GitHub and will be made public upon publication.

[13] The full template can be found in §D.

and political bias. As we increase the amount of instructions provided in our prompt, we aim to use prompting to guide the models towards generating a more favorable answer.

- *(Zero-shot) Roleplay*: The prompt asks the model to role play as a text classification model, which understands linguistic nuances and is well-versed with the political discourse/scenario in the United Kingdom since 2018.
- *(Zero-shot) Content*: Additional to the roleplaying, we also provide definitions of sexist and political attitudes to the model (the definitions we use in our work).
- *Zero-shot*[14]: Alongside the definitions, we provide information on the linguistic cues we want the model to be aware of, such as emoticons, quotations, etc.
- *Few-shot*: To guide the model further, we add some examples to the previous prompt, that can potentially remove any existing biases from the generated language.

## 3 RESULTS AND ANALYSIS

In this section, we present with two sides of analysis for the predictive capabilities of LLMs in political discourse. The first subsection majorly focuses on the performance evaluation and confidence estimation – both of which provides a good idea on the trend of occurrences of sexism in political discourse, as well as tests the reliability of predictions by the LLMs, when provided with various categories of prompting. While the second subsection focuses solely on the qualitative analysis from our methodology and results.

### 3.1 QUANTITATIVE ANALYSIS

| Prompt Category | ChatGPT | | | | Flan-T5 | | | | Alpaca | | | | Vicuna | | | | Mistral | | | |
|---|---|---|---|---|---|---|---|---|---|---|---|---|---|---|---|---|---|---|---|---|
| | R | P | F1 | A | R | P | F1 | A | R | P | F1 | A | R | P | F1 | A | R | P | F | A |
| Roleplaying | 64.78 | 54.68 | 44.36 | 52.95 | **66.11** | **59.45** | **61.13** | **83.19** | 47.68 | 49.08 | 28.14 | 30.71 | 51.10 | 50.35 | 37.51 | 45.25 | 48.13 | 48.46 | 48.25 | 81.93 |
| Content only | **70.04** | 58.02 | 57.91 | 75.33 | 64.80 | **59.28** | **60.85** | **83.82** | 51.24 | 51.05 | 17.65 | 17.67 | 52.70 | 50.92 | 45.84 | 62.45 | 49.60 | 49.77 | 49.06 | 77.85 |
| Zero-shot learning | **68.37** | 56.14 | 51.97 | 65.51 | 61.92 | **59** | **60.08** | **85.31** | 50.59 | 50.32 | 22.65 | 23.25 | 54.19 | 51.47 | 47.12 | 64.41 | 49.79 | 49.87 | 49.37 | 78.95 |
| Few-shot learning | **72.86** | 57.88 | 55.25 | 69.21 | 62.99 | **61.50** | **62.17** | **87.27** | 50.30 | 50.10 | 34.87 | 40.77 | 48.24 | 48.39 | 48.31 | 82.88 | 49.52 | 49.26 | 49.22 | 86.72 |

Table 3: This table documents the performance metrics we used in our study: macro-Recall (**R**), macro-Precision (**P**), macro-F1 (**F1**) and accuracy (**A**) scores for each models and their prompt categories. All measures are recorded in percentage(%). The best scores of each metrics are highlighted in bold and underlined.

#### 3.1.1 PERFORMANCE EVALUATION

As we see in Table 3, most of the models seem to under-perform in detecting sexism against non-sexist texts, in our political discourse through prompt engineering. The disparity in performance between LLMs can be attributed to the specific tuning conducted to optimize their pre-trained versions for chat compatibility (Kumarage et al., 2024). We report our experimental results using macro-averaged scores of multiple classification evaluation metrics (accuracy, recall, precision, F1-score), given the imbalance in the dataset. Ideally, the recall score is favourable over other evaluation metrics since recall is the measure of the ability of a model to define the true positive sexist speech. Having a lower recall would suggest that there are larger linguistic patterns that the model would not be able to detect (Warner & Hirschberg, 2012). Flan-T5 performed the best in the roleplaying prompt category, while ChatGPT performed better in the other prompts. However, it is important that the model shows good overall performance in most of the metrics, to be considered ideal for any task at hand, such as the sexism classification task in our case. Given that Flan-T5 showed an impressive overall performance in spite of being an open-source model, much higher than the other metrics, we decided to perform further checks on their performance with respect to their generated text at each instances. In our research, we prefer open-source models as they enable researcher to experiment and come up with approaches to improve it further, whereby making them the ideal choices for conducting research in a cost-effective way.

#### 3.1.2 CONFIDENCE ESTIMATION

Given that LLMs are prone to hallucinations (Huang et al., 2023) and lack consistency (Elazar et al., 2021), it is essential that uncertainty measures are used to improve quality assessments by estimating the confidence of the models' prediction. Therefore, in this section, we explore confidence estimation of the LLMs using relative entropy (an entropy-based confidence estimation which builds

---

[14]*Note: All the first three prompt categories are instances of zero-shot learning. They differ in the level of instructions fed to the model, with increasing order of information provided across the categories from the top to bottom. For the sake of simplicity, we name them as 'roleplay', 'content' and 'zero-shot' accordingly.*

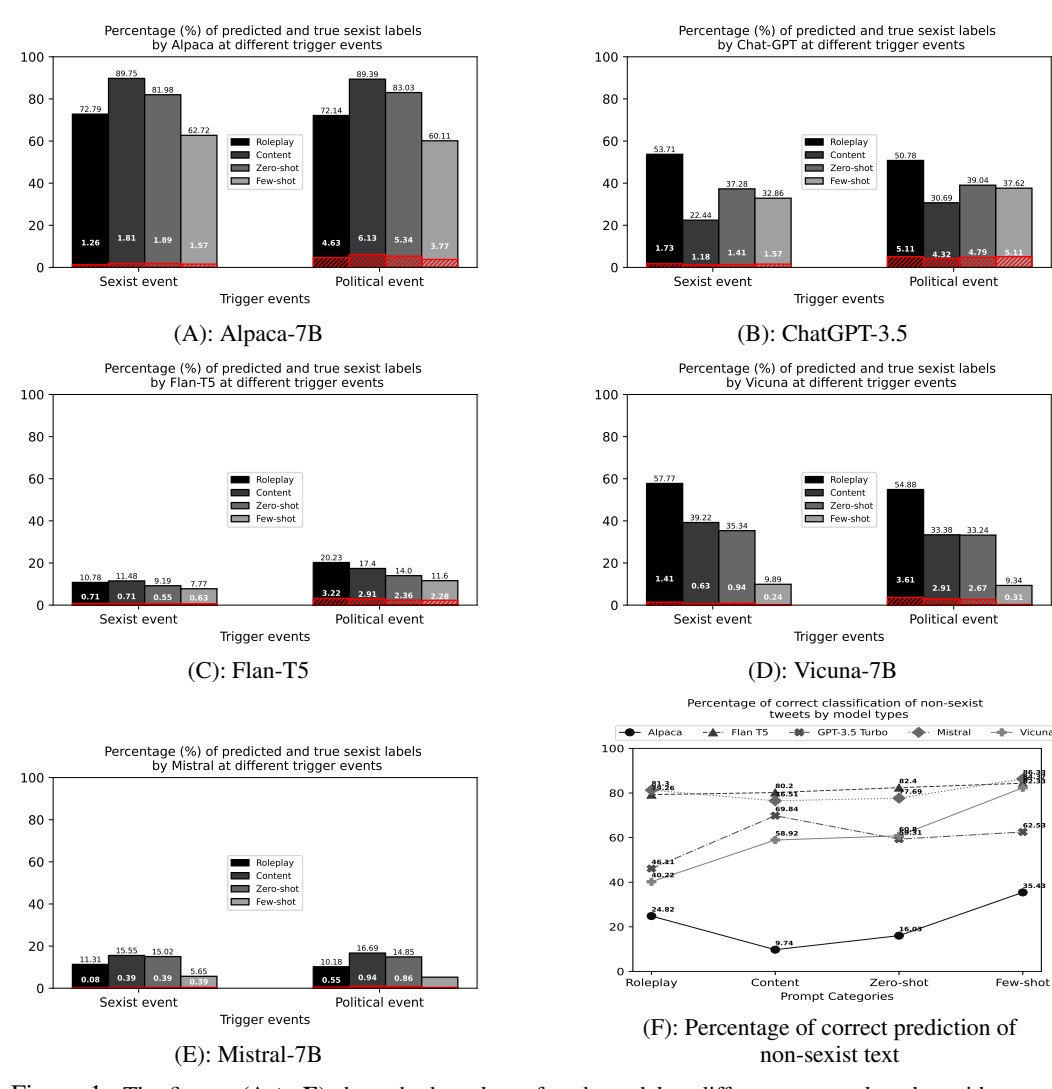

Figure 1: The figures **(A to E)** show the bar plots of each model at different context lengths, with respect to the trigger events. Each bar plot depicts the proportion of total entries that were considered 'sexist' by the model, with each color indicating their respective prompt type, as per the trigger events. Simultaneously, the shaded bar plots in red, in the same bars depict the proportion of the those predicted entries that were actually 'sexist' (i.e., true=predicted='sexist'). The figure **F** indicate the number of correct 'non-sexist' predictions by each model, with each prompt type, irrespective of the trigger event types.

on probabilistic tools for uncertainty estimation) to determine if the outputs they generate should be considered based on their uncertainty levels. Following our previous section and inspired from Frantar et al. (2022), we check the confidence estimation in the generated text output using Flan-T5. The intention is to see how well the confidence score can predict the correctness of the LLMs. While most of the previous research are either performed on closed-source model like the GPT-series, or depend on semantic equivalence like Kuhn et al. (2023) to capture the meaning from the text instead of focusing on the tokens, we decided to not use either of them because of the mismatch in the model type and our consideration of the output text for the classification task, rather than for text generation task. While text classification in prompt engineering is also a form of text generation, it is more focused on classifying the said text as one of the two labels provided to them in any classification levels (in our case, it is a binary classification task: 'sexist', 'not sexist'). Similar to the semantic entropy proposed by Bansal & Desai (2023), we propose calculating relative entropy for generation-like task to perform confidence estimation of the LLM. For an input sequence $X$ containing our prompt at any level of instruction, it produces an output $Y \in \{Y_1, Y_2\}$ corresponding to the categories of classification. When iterated through different seed values, various instances of output $Y = (y_1, y_2, \ldots, y_n)$ are produced. At each instance of the output, the proba-

bility of predicting the output token $y_i$, $p(y_i|X)$ is determined using a softmax with temperature $\gamma$: $p(y_i|X) = \exp(z_i/\gamma)/\sum_{j=1}^{N}\exp(z_j/\gamma)$ where $z_i$ is the logit score for token $y_i$. Prior studies (such as Chen & Ding (2023)) indicate that temperature ($\gamma$) scaling impacts creativity in a model, bringing in instability and producing invalid answers. Therefore, we prompt the LLM through three unique seed values to produce $n = 3$ iterations of prediction at a temperature of 0.0 (as higher value of temperature introduces randomness while lower value makes the output more deterministic by favoring the most probable words i.e., our class categories), along with their log probabilities $p(y_i|X)$ of the generated token $y_i$ (see Algorithm 1).

---

**Algorithm 1** Relative Entropy

---

**Require:** $X$ (Prompt) $\leftarrow$ **input**
   $Y \leftarrow \{y_1, y_2, y_3, \ldots, y_n\}$                ▷ Sampled label predictions
**Ensure:** $: Y \in \{Y_1, Y_2\}$               ▷ Label choices must be binary (our case)
  **for** $y_i \in Y$ **do**
     $c_i \leftarrow \text{Mean}(\log(p(y_i|X)))$        ▷ Sequence log-probability of each output
  **end for**
  $H \leftarrow \Sigma_{i=1}^{n} 1/2(c_i \cdot \log_2(c_i/c_{i+1})) + 1/2(c_{i+1} \cdot \log_2(c_{i+1}/c_i))$    ▷ Jensen-Shannon divergence
**Ensure:** $:$ Divergence score should be calculated among the iterations for the same prompt category
  $C \leftarrow 1 - H$                 ▷ Current prediction confidence

---

Inspired from Guerreiro et al. (2023), we explore the *uncertainty quantification* of the LLM by calculating the sequence-level log probability (Seq-logprob) of each output are to be collected for every instance after each iterations, to convert the log probabilities to a more easily interpretive scale of 0-1 (or 0-100%). Seq-logprob is the length-normalized sequence log-probability, denoted by $\frac{1}{L}\sum_{k=1}^{L}\log p(y_k|y_{<k}, X)$, where $L$ denotes the number of tokens in the generated output. Upon collection, we quantify the difference between the probability distributions from each iteration (two at a time), for each entry by the model, using Jensen-Shannon Divergence (JSD) score $H$. In probability theory and statistics, JSD is a method of measuring the similarity between two probability distributions (as explained in Algorithm 1). It is a symmetric and smoothed version of the Kullback–Leibler divergence $KL(P||Q)$ denoted by $\sum_i P(i) * \log(P(i)/Q(i))$, where $P$ and $Q$ are the target and predicted probability distributions respectively. It is denoted as $JSD(P||Q) = \frac{1}{2}KL(P||M) + \frac{1}{2}KL(Q||M)$ where the value of $M$ is calculated as the average of $P$ and $Q$, i.e., $M = (P + Q)/2$. Given two distributions from multiple iterations with more than one output expectations (i.e., classes), we group the same (or similar, i.e. indicating the same output as the class itself, even though we would ideally want the LLMs to generate the tokens representing the classes) output per instance together and calculate the average confidence per class. In case the model generates different outputs at each iteration, we advise five iterations or more. Otherwise, three iterations should be sufficient. Lesser the divergence between the two distributions of similar output from the iterations, more confident the model is about their prediction (whichever has a higher confidence score), consequently indicating the understandability of the model. After the three iterations, our model Flan-T5 demonstrated an impressive score of $\approx 1.0$ (100%) confidence score (see Table 4) with the same predicted labels in all the entries, across all the iterations. That leaves us with very little doubt on the confidence of the model in its predictions, as the divergence (difference) between the probabilities remain the same, regardless of the number of iterations. Following the impressive performance and confidence from Flan-T5, we now test if the Seq-logprob are indicative of correct predictions by the model.

|  | **Roleplaying** | **Content** | **Zero-shot** | **Few-shot** |
|---|---|---|---|---|
| **Confidence** | 0.99 $\pm\,1.50e-12$ | 0.99 $\pm\,1.12e-12$ | 0.99 $\pm\,8.30e-13$ | 0.99 $\pm\,6.72e-13$ |

Table 4: This table documents confidence of the Flan-T5 model across the different prompt types.

| Correct Predictions considering base as (truth == prediction) | *based on* mean of log probabilities of a particular iteration | | | |
|---|---|---|---|---|
|  | Roleplaying | Content | Zero-shot | Few-shot |
| Correlation | -0.47081 | -0.4133 | -0.4491 | -0.4341 |
| p-value | $3.67e-71$**** | $1.16e-53$**** | $3.55e-64$**** | $1.38e-59$**** |

Table 5: This table documents the results from the Point Biserial Correlation.

From Table 4, we see that the models generally show high confidence in all of their predictions (irrespective of correctness). Therefore, we use Point Biserial correlation to determine the relationship or the strength of association of the misclassifications by the model (variable indicated as *True* if mispredicted by the model, else *False* in the dataset) and the Seq-logprob scores – i.e., to check

if lower confidence can be associated with possible misclassification, alongside all of the prompt categories, as shown in Table 5. At 0.05 significance level, the correlation coefficient for all the prompt types lie within the range of $-0.41$ to $-0.47$, indicating a moderate negative correlation, which is also statistically significant (indicated by the p-values), between the log probabilities and the correct predictions. Thus, it implies that predictions having a higher mean of log probabilities of the generated tokens, tend to be correctly predicted on the overall dataset (which would indicate an anomaly) and predictions having lower mean of log probabilities tend to be incorrectly predicted (also an anomaly). For example, the overall Seq-logprob of the correct predictions are seen to be $0.9743(\pm0.0273)$ in roleplay prompting, which is a few points higher than incorrect predictions of sexist entries ($0.9545\pm0.0295$) and that of non-sexist entries ($0.9252\pm0.0329$) for the same prompt category. This further proves the effectiveness of using Seq-logprob in LLM prompt engineering for detecting possible mislabeling (or mis-annotation by the respective model) in any datasets.

## 3.2 QUALITATIVE ANALYSIS

### 3.2.1 SENSITIVITY TO PROMPT DESIGN

We find that the generated output is sensitive to prompt designs, and is often difficult to infer if the results were a result of prompt designing. We used multiple manually written discrete prompt templates to test, and a set of language model targets for the classification task to compare responses from each model. While Webson & Pavlick (2021) found that the choice of the target words in models usually override the meaning of the overall prompts, they also agree that learning from instructions is an important research direction. Therefore, given these limitations, we use this research to investigate the models' understanding of the prompts and predicting sexism in texts accordingly.

### 3.2.2 ERROR ANALYSIS

**Model Predictions** Figure 1 demonstrate the proportion of sexist predictions in each model, suggesting the models' varied proneness in labeling a text as sexist. Yet, the accurate detection of sexism by the models (shown in proportion as shaded red blocks in the bar plots) indicate that most of the models are prone to mislabeling, and have a high false positive rate. Given the huge imbalance in our dataset, our expectations from this LLMs, when fed with descriptive prompts, were that they would improve in their performance of predicting sexism. While Alpaca-7B predicts an alarming number of texts as sexist (consistently about 60% of the entries) even when fed with more instructions/prompts, it is only about right for less than 2% in all the prompt categories. This poses a challenge in its usage for prediction of online sexism, especially in political contexts. While Vicuna and ChatGPT too have shown to have biased judgments on sexist content, given that they labeled nearly 50% of their data as sexist, both predict lesser sexism in further prompt categories. The risk of false positives (i.e., false accusations) is a risk in automated methods as it may lead to over-blocking or removal of harmless content from social media with little moderator interventions (Markov & Daelemans, 2021). This is especially a problem in political discourse as the wordings of the text themselves, though seemingly linear composed, leave a lot of room for interpretations by the readers (Van Dijk, 2002). Hence, it is imperative that we reduce the false positives since political engagements in online platforms promote greater political participation and increase in the size of online discussion networks (Valenzuela et al., 2011), if constructive and civil, is needed within a democratic society (Johnson & Johnson, 2000). We see that, of all the models, Mistral-7B and Flan-T5 performs the best in correctly detecting non-sexist texts (Figures 1F and A3) which improves for both as the prompt categories become more instructive. Previous work indicate Mistral's heightened capability in identifying sexism with few-shot learning (Siino & Tinnirello, 2024). Yet, in the political discourse, Flan-T5 shows a preferable capability in identifying *both* sexist and non-sexist texts, thereby reducing false positives and false negatives (i.e., missed accusations).

**Model *Mis*predictions** Language can be a potent vehicle for subtle sexism (and even socially acceptable), while also a driver to reinforcing equality (Chew & Kelley-Chew, 2007). It is therefore always difficult to interpret non-sexist/sexist texts when the models are focused on certain keywords (Dutta et al., 2024). Gothreau et al. (2022) recognize all the forms of sexism which exist in political discourse- namely hostile, benevolent, modern, and an implicit form of sexism that exists to capture sexist attitudes that may exist outside of one's conscious awareness. Such texts could be in the form of insinuations, sarcasm, jokes or references to fictional characters from books or movies, yet are hard for the models to understand. We list a few examples which most of the models mis-predicted and briefly discuss the possible reasons that could have contributed to the model misclassifications.

e.g., *"@MENTION1 Still can't see your name without chuckling at you flashing MENTION2 your Ginger Growler"* **Label (L)**: 'sexist'; **Prediction (P)**: 'not sexist'

e.g., *"Notice MENTION3 was one of the first to condemn this? Could it be that he looked far worse than MENTION4, that he is so easily distracted"* **L**: 'sexist'; **P**: 'not sexist'

Even though LLMs are thought to be more context-aware, any subtle indications of sexist attitudes in texts are not captured by the models, regardless of the speaker's apparent political position.

e.g., *"@MENTION1 It's in the Mirror, there's no reality in that rag!"* **L**: 'not sexist'; **P**: 'sexist'

Similarly entries that are undoubtedly non-sexist are predicted to be sexist because they contain strong and explicit language and slangs, even though the overall meaning of the sentence are imply harmless intent.

e.g., *"@MENTION5 and who fuck cares about a bunch of weekend warriors. The only opportunity they have to wear uniforms like that is so they can take pictures like this"* **L**: 'not sexist'; **P**: 'sexist'

e.g., *"@MENTION6 She wasn't sacked. Stop lying"* **L**: 'not sexist'; **P**: 'sexist'

More so, when the entries contain partisan bias or express discontent in the performance of politicians, the model simply predict them as sexist.

e.g., *"MENTION7 has brought the sensationally low-competence, low-calibre MENTION8 back as [POSITION]. What has she ever achieved, bar "annoying all the right people"?"* **L**: 'not sexist'; **P**: 'sexist'

e.g., *"#ResignMENTION9 has done more to damage womens rights than any male politician #dontvotesnp"* **L**: 'not sexist'; **P**: 'sexist'

Overall, we see that the LLMs generalize on the explicit or obvious forms of online sexism, while missing the more subtle and implicit forms. And yet, when provided with more instructions in the form of prompts (including examples), their performance do not necessarily improve.

## 4 CONCLUSIONS

In this paper, we first define sexism in a political discourse and identify the trigger events of both sexist and political nature that causes high-activity in social media, potentially leading up to sexist discussions around the female politicians. We further investigate the performance of LLMs through prompt engineering to test their efficiency in capturing linguistic nuances which are very typical to the political discourse (§3.1.1 & §3.2.2 for **RQ1**). This indicate that prompting categories in annotation task may not be as important in detection cases as we had previously considered in this study, indicating that it may only be useful when considering the difference between these prompting categories in terms of the label noise. Consequently, we propose an improved algorithm to check the confidence of the model in their predictions, and indicate if their confidence in turns impacts on their prediction capability (§3.1.2 for **RQ2**). This algorithm is aimed to improve on the existing uncertainty quantification through simple implementation, and can be replicated across any generative models for performing classification tasks. While we find positive results in our approach and hypothesis testing to assert our observations, our evaluation results on the LLMs show their 'underperformance' in the sexism classification task, as compared to their competitive results for the same task (such as Morbidoni et al. (2023)) in datasets from other domains. Our qualitative analysis further confirms the drawbacks of using LLMs as much of the work goes into designing the prompts and mitigating their inherent bias (§3.2 for **RQ3**). It is therefore essential that we reach the stage in research where the bias in LLMs could be controlled and mitigated further, before we use them to detect online sexism in a polarised discourse, such as politics. However, we also acknowledge the potential of LLMs in improving on their performance if they are trained with more representative examples consisting of the subtle and implicit forms of sexism, alongside instruction tuning. To improve evaluation, models require fine-tuning with labeled entries from political discourse to improve its understandability, which is an expensive process. Till then, we can only trust LLMs in predicting the more conventional forms of online sexism. At this point when more researchers are turning to the capabilities of LLMs in annotation tasks, our research insights can provide sufficient information on their performance in detecting online sexism and help researchers make informed decisions regarding incorporating LLMs as annotators. We hope that this work will promote further research in enhancing annotation performance of the LLMs to bring it closer to the quality of human-generated labels before they are used as human replacements for annotations in domain-specific NLP tasks.

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

## A   ANNOTATION INFORMATION

We had collected annotations in two phases, where we had two annotators label 900 of the tweets in the pilot phase, with one annotator per tweet (this is for the majority of the tweets). In the next phase, we experimented with the remaining number of tweets, having three annotators per tweet, among seven annotators (including the two annotators from before). Any other socio-demographic information of the participants were not collected as it did not seem relevant for this study. We had provided them with the annotation guidelines[15] and conducted regular meetings to discuss their task and the purpose of this work. While prior studies have usually performed majority voting with multiple annotators per instance to mitigate voting bias, there are studies (such as Fleisig et al. (2024); Basile et al. (2021); Khurana et al. (2024)) which have also explored why majority voting may not be ideal in subjective tasks [like sexism]. In such tasks, inferring truth from labels are still considered an open problem (Zheng et al., 2017). Therefore, to promote subjectivity in the annotations among the annotation pool, we decided to follow minority voting scheme of the annotations in the second phase. Through this voting scheme, we believe that it provides preference to every annotator viewpoints, and promotes subjectivity in tasks such as sexism.

## B   LIMITATIONS

### B.1   RELATION BETWEEN TRIGGER EVENTS AND CHOICE OF MPS

The choice of MPs was based on the volume of tweets collected on that particular timestamp, as we mention in §2.1.1, and we describe each of the trigger event types based on an offline event, calling it sexist or political based on the definition we use in our research. However, we acknowledge that the MPs are synonymous to their party affiliation and identity or social attributes (such as their race, professional position, approach, etc.) and the sexism they receive could be a result of any/all of that (or not). Additionally, it also depends on the trigger event chosen and in what way it impacts the MPs in question. Therefore, we cannot claim that the choice of MPs for our study is ideal to explore the different forms of sexism women in politics face in online spaces. However, due to the lack of resources, we had to make a conscious decision to either go for a study which explore the trust on LLMs' judgement in detecting the forms of sexism in online political discourse through multiple intersectional components, and the trust when we have sexism as a binary quantitative measure while consequently measuring their confidence in judgement. We chose to go with the later due to the availability of the data.

### B.2   ONE HUMAN ANNOTATOR USED PER INSTANCE

Like we mentioned in §2.1.3, seven experts were assigned to annotate the data as 'sexist' or 'not sexist' for this project. Due to limited resources, we had assigned only one annotator per entry for most of the cases. Considering the subjective nature of the topic i.e. sexism, and the background of the annotators, those labels were assumed to be ground truth without further inspection. We understand that this could be taken as a limitation in obtaining a robust dataset, but the intention for this research was to demonstrate the capability of LLMs in detecting sexism in online political discourse. Therefore, we anticipate future works extending from our research to test the trustworthiness of the LLMs with a more robust dataset, and form informed decisions in further assigning them as annotators.

### B.3   CONFIDENCE ESTIMATION TESTED ONLY IN THE BEST MODEL

While a comparative analysis of the confidence estimation among the LLMs would have been appreciated, we felt that it was beyond the scope of the current research. The intention was to demonstrate the effectiveness of the proposed algorithm, regardless of the LLM used, as all of them operate in the same way. If we capture their uncertainty on every output and consequently calculate their confidence through the multiple distributions (obtained through the iterations), coupled with a good performance in the said task, we may be willing to use them for performing annotation tasks.

### B.4   MEASUREMENT OF INTER-OUTPUT SIMILARITY OF THE MODEL

Since our best model Flan-T5 generated output tokens same as the class categories i.e., 'sexist' or 'not sexist' (like we mentioned in §3.1.2), we did not require to check for the inter-output similarities for the same instance, between the multiple iterations of the LLM. Though this is an important step, this research does not provide an evaluation metric that can efficiently recognize similar outputs

---

[15]To be published in Github repository and briefly discussed in §2.1.3.

from the LLM, if they differ (significantly) from either of the classes. Similarity evaluation metrics such as BERTSCore (Zhang* et al., 2020) is usually used to compute the similarity score between the two texts (candidate text and reference text), which computes similarity using contextualized token embeddings. Our initial analysis with BERTScore showed flawed outcome as it computed a much higher score at the event of misclassification ($\approx$0.9). This could be a result of the class categories which are similar when matched with each of the tokens. Alternatively, one may use generative LLMs itself to evaluate the similarity of the generated output to any of the classes, to indicate if the outputs are same or different across the multiple iterations.

### B.5 EXPLORATION OF POLITICAL BIAS IN POLITICAL DISCOURSE

From our observation, it may be possible that the models' performance were influenced by their political bias. Previous works (such as Motoki et al. (2024)) have found political bias in ChatGPT, and studies have found that these biases stem from political opinions in training data (Santurkar et al., 2023). Though most of such works base show the LLMs' political bias based on USA politics, we believe that the same may be true for the politics in the UK. Therefore, exploring political bias of LLMs in UK politics would be a good future direction, although not explored in the current work.

## C DATA SPECIFICS

### C.1 EMOTION ANALYSIS OF CONVERSATIONS DIRECTLY MENTIONING THE TARGETED MPs

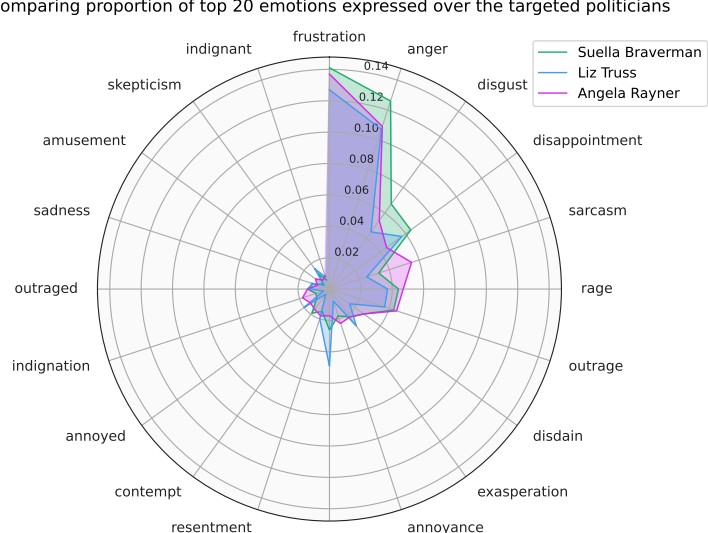

Figure A1: Polar plot depicting the top 20 overall general sentiment towards the targeted female MPs. The value of each emotion is calculated with respect to all the emotions shown towards them, hence a proportionate value.

As mentioned in §2.1.1, we selected four trigger events based on the online activity centering the events around those selected periods. While the data was collected with mentions of all the 38 female MPs, we had sampled the data further based on the engagement metrics (see §2.1.2) around that period. We see that out of the 1273 tweets collected, 682 of the tweets come with the direct mentions of any or all of the targeted female MPs i.e., targets of the trigger events. To understand the possible source of misclassification of the LLMs, we attempted to extract the emotions out of the corresponding instances to see the public sentiment directed at them, at the event of these trigger events. For the emotion extraction task, we used Llama 3[16] to prompt the model in recognizing the emotions expressed by the speakers, since del Arco et al. (2024) demonstrated the LLM's capability in calibrating emotions along a text. As shown in Figure A1, 'frustration' and 'anger' dominated over all other emotions for all the three candidates. Aside from those, for the Conservative candidates the emotions of disgust and disappointment seem to prevail for Suella Braverman, while disappointment and concern for Liz Truss. Given that the events centering around both were po-

---
[16] https://huggingface.co/meta-llama/Llama-3.1-8B

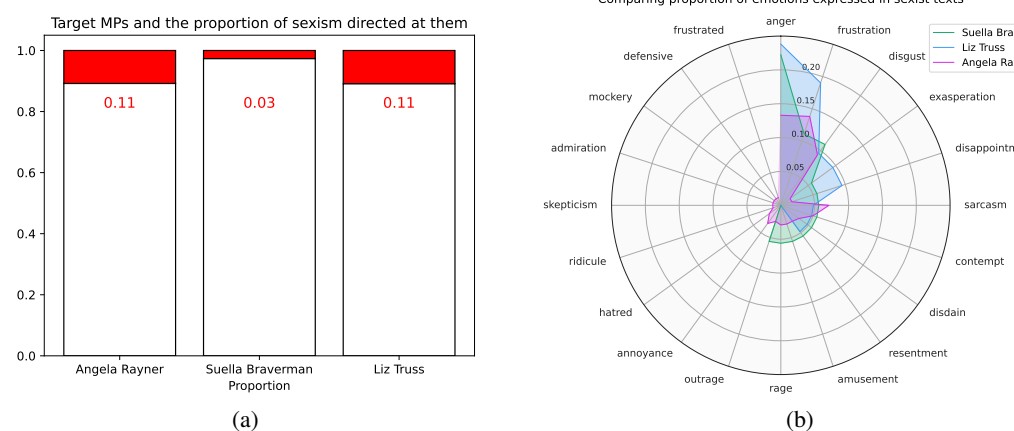

(a)  (b)

Figure A2: Plots depicting the share of sexist texts directed at the targeted politicians. Figure (a) shows the bar plot with the proportion of sexist comments marked in red, Figure (b) illustrates a polar plot of the most common emotions of the speaker from their sexist texts.

litical, these emotions (which are highly negative) align with the general public sentiment towards politicians of the UK (Commission, 2023).

Given that the LLMs particularly fell short in identifying sexism in the political discourse (as we see in §3.1.1), we decided to analyze the emotions that were attributed to the sexist texts directed at the targeted MPs of the trigger events. Figure A2a demonstrates the proportion of texts (which had the mentions of the targeted MPs in it) that were sexist. Consistent with the previous figure, we see in Figure A2b that anger still dominates over the other emotions in sexist texts, though much more than frustration for Liz Truss. However, the level of anger and frustration is much reduced and marginally replaced with disgust and sarcasm for Angela Rayner. Even though the trigger events are different in their types and periods of discussion, the overall emotions are mostly consistent throughout. Therefore, it implies that in political discourse, it is very difficult to detect sexism through the tone and phrasing alone, if one is not aware of the context. While we discussed the trustworthiness of LLMs in §3.2, our emotion analysis further proves how the nature of political discourse might be a reason why the LLMs under-perform in recognizing sexism (like we see in §3.1.1).

## C.2 DOES PERPETRATION OF SEXISM DEPEND ON TRIGGER EVENTS?

Though we see a noticeable difference in the number of sexist content between the two types of trigger events (see Table 2), we test if the difference between the groups are significant. Kalyanam et al. (2016) observe that activities triggered by real-world news events follows a similar pattern to that observed in other types of collective reactions to events, which is displayed by periods of intense activity as well as periods of inactivity.

| Label | *grouped by* trigger event types | | |
|---|---|---|---|
| type | p-value | Effect size | Magnitude |
| Sexist *vs* non-sexist | 0.000011**** | 0.123 | Small |

Table A1: This table documents the results from the non-parametric Wilcoxon signed rank test.

To assert our assumptions that the differences are indeed significant enough between the two groups of trigger types, such that the occurrence of sexist conversations are more at the event of sexist trigger events, we use the non-parametric Wilcoxon signed rank test, as shown in Table A1. At 0.05 significance level, we conclude that there is a significant difference between the two kinds of entry types, given the trigger events, even though the magnitude and effect size is small (considering the number of instances taken for each). Therefore, this leaves us to believe that between the two groups of samples (trigger type events), the population for each entry type (i.e., sexist or not) is non-identical in nature.

## D PROMPT INSTRUCTIONS

| Prompt Category | Prompt Template (with increasing instruction on the context, content and phrasing) |
|---|---|
| Roleplay | *"You are an expert in understanding slight linguistic nuances in the text, even when presented with texts that lack enough context. You are well-versed with the political discourse/scenario in the United Kingdom since 2018, especially in social media platforms like Twitter.* + 2 options (sexist, not sexist) to choose labels from |
| Content | ... + Instructions for understanding 'content' through linguistic cues + 2 options to choose labels from |
| Zero-shot (Radford et al., 2019) | ... + Instructions for understanding 'content', 'context' and 'phrasing' through linguistic cues + 2 options to choose labels from |
| Few-shot (Brown et al., 2020) | ... + Instructions for understanding 'content', 'context' and 'phrasing' through linguistic cues + 2 options to choose labels from + [representative examples *(n=5)* for each label] |

(a)

| Instruction Type | About | Linguistic Information provided |
|---|---|---|
| Context | Regarding the political or non-political incident in question | The context is mostly political and it consists of texts that revolve around discussions of policies, government actions, ideologies, elections, etc. These texts would aim to engage with societal issues, power dynamics, and decision-making processes within the realm of public affairs. Pertaining to the practice and theory of influencing other people on a civic or individual level, often concerning government or public affairs. Reference to any of the target's current or former political and/or behavioral activity. This could be an implicit indication in the text, or a direct implication through mentions of their position on a certain topic. A typical political text could have strong language, a harsh tone and slurs; and question the political standing and political opinion of the target (usually indicated by mention of policies or government strategies) or the political position the target holds. Yet it should not undermine the intelligence of the target. Texts could be mocking female perspectives from female politicians, minimize their political contributions and undermine their achievements. It can also question their commitments to public office by implicating that they should focus more on their family commitments, and their political performance being compared to their capability in familial setting. They may also publish appearance-centric criticism of the female politicians, unlike their male counterparts. They tone could be ironic and exaggerated, and often in the guise of humour. |
| Content | Regarding what the speaker believes | A text may be sexist if the speaker shows a prescriptive set of behaviors or qualities, that women (and men) are supposed to exhibit in order to conform to traditional gender roles. This could be texts formulating a descriptive set of properties that supposedly differentiates the two genders and expressed through explicit or implicit comparisons and perpetuating gender-based stereotypes. Aside from acknowledging the inequalities, these texts could be endorsing or justifying them in a non-flattering manner. This may contain texts stating that there are no inequalities between men and women (any more) and/or that are opposing feminism. They might possess views which indicate women are not competent adults, or women having favourable traits that men stereotypically lack. For example, the speaker may express sexist attitudes towards gender inequality, either endorsing it (e.g. "some jobs are best left to men"), or antagonizing it (e.g. "the pay gap between genders does not exist, feminists should stop complaining"). Also, the speaker may express stereotypes (how genders are traditionally seen and compared to each other) and behavioral expectations (how individuals of a gender should behave according to traditional views). Sexism may also include positive stereotypes (e.g. "women are the best home cooks"), or target men (e.g., "men should not cry"). |
| Phrasing | Regarding the speaker's choice of words | Texts may be sexist simply because of how the speaker phrases it–independently from what general beliefs or attitudes the speaker holds. A message is sexist, for example, when it contains attacks, foul language, or derogatory depictions directed towards individuals because of their gender, e.g. by means of name-calling ("you bitch"), attacks ("I'm going to kick her back to the kitchen"), objectification ("She's stupid but I'd still do her"), inflammatory messages ("burn all women"). However, just because a message is aggressive or uses offensive language does not mean that it is sexist. |

(b)

Table A2: In Table (a), we provide the general prompt templates across all the models taken for this study. With each prompt from top to bottom, we increase the amount of instructions provided. In Table (b), the prompts are further elaborated based on the linguistic cues of each instruction types.

# E   FURTHER INSPECTION ON THE PERFORMANCE OF THE LLMS

## E.1   SEXISM PREDICTION ACCURACY OF LLMS BY THE INDIVIDUAL MPS

LLMs have demonstrated gender bias, amplifying stereotypes associated with the female individuals, more than those associated with male individuals Kotek et al. (2023). Hence, a LLMs' specificity i.e., true negative rate, or the proportion of actual negative (sexist) cases that are correctly identified as such by the model, might help us test their gender bias. While we determined the emotions of the texts directly mentioning the targeted MPs in §C.1, we also measure the specificity of LLMs by the texts having direct mentions of each MPs.

Figure A3 shows the line plots of the specificity score for all the models across the prompt types, by each target MPs. Though all the LLMs generally show the same trend for all three cases, the models which marginally vary more than the others are ChatGPT and Flan-T5. Though previous

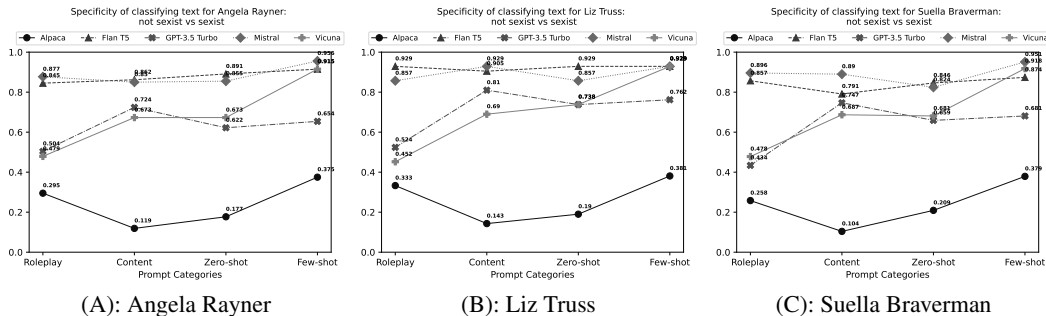

(A): Angela Rayner  (B): Liz Truss  (C): Suella Braverman

Figure A3: Plots depicting the specificity by a single LLM across the different target female politicians used in our research. A lower score means that it is more likely that the respective LLM generates a higher number of false positives and may incorrectly identify sexism in a text when it is not present. Conversely , higher value would mean possibility of fewer false positive values, and hence more preferable.

works have found bias in the LLMs, including political bias, it is not evident if that is the reason for inconsistencies in the models' specificity. When we compare the scores of the evaluation metrics across targeted MPs, the differences in performance could be attributed to more than one bias – a product of the target's intersectional identities.

