# OpenReview forum: "We are confident in trusting Language Models for annotating online sexism in political discourse, but are they good?"
_ICLR.cc/2025/Conference — Submitted to ICLR 2025_

### Official Review · Reviewer_CaGn · 2024-11-03

**Soundness:** 3
**Presentation:** 3
**Contribution:** 3
**Rating:** 5
**Confidence:** 5

**Summary:**

This paper brings a focus on the task of detecting sexism specifically from political discourse. It includes an assessment of existing LLMs through prompt engineering when it comes to this task and then propose a new algorithm for capturing the confidence of the LLM
predictions in classification tasks. Their methodology illustrates trigger events in text that provoke sexism. Nonetheless they also claim that it provides no clear advantage when predicting sexism.

**Strengths:**

The model covers a thorough analysis of sexism detection on political discourse online using state of the art Large Language Models. For such a niche task, prompt engineering seems to be the right approach but they also put a detailed analysis into the robustness of the prompts they use as well as the calculation of a confidence score based on the outputs yielded by LLMs.

**Weaknesses:**

While this work addresses an important issue in language i.e. sexism detection and also provides a thorough picture of state-of-the-art LLMs when it comes to solving it, the overall novelty of this paper is low. It does not provide any specific understanding of what drives sexism within political narratives and or solutions of how models can be trained to capture such semantic cues in text.

**Questions:**

The central thesis of this paper is understanding and detecting sexism in political discourse, yet it fails to specifically highlight what about political discourse require special investigation than regular text for sexism detection.

---

> ### Author Response · Authors · 2024-11-18
>
> We thank the reviewer for their feedback and for taking the time to go through our paper. There are some points we felt would provide more clarity for the reviewer.
>
> > It does not provide any specific understanding of what drives sexism within political narratives and or solutions of how models can be trained to capture such semantic cues in text.
>
> We understand that exploring sexism within political narratives can be done from this research, but that is beyond the scope of this current research given that we did not aim to analyse the biasness in the model. To study the drivers of sexism in political narratives, we would require a longitudinal study and consideration of a wider range of political figures along with their intersectional attributes which could contribute to sexism. Through the different prompt categories, we aim to provide linguistic/semantic cues (details in the appendix) to the model to help them generalize the text at a more localised level.
>
> > The central thesis of this paper is understanding and detecting sexism in political discourse, yet it fails to specifically highlight what about political discourse require special investigation than regular text for sexism detection.
>
> We realized from the reviews that we should elaborate more on why understanding and detecting sexism in political discourse is required. We briefly discuss that in the appendix section, but we aim to elaborate on that in the revised version.
>
> We thank the reviewer again for their feedback.

---

### Official Review · Reviewer_ETQd · 2024-11-03

**Soundness:** 2
**Presentation:** 2
**Contribution:** 2
**Rating:** 6
**Confidence:** 3

**Summary:**

The paper investigates how efficient are LLMs in predicting sexism in online discourse. The experiments are conducted on a twitter dataset conducted as part of the study which includes tweets from 22 with location as UK. They use 5 different LLMs and 4 different prompt types to evaluate the sexism prediction on the dataset. In addition, the also introduce a metric for predicting confidence of the models in their predictions using an uncertainty estimation.

**Strengths:**

- The paper uses an extremely imbalanced dataset to predict sexism in online political discourse.However, this is similar to real-world scenario where majority of the texts are non-sexist, making the research question quite interesting.
- In addition to model comparison, they also perform error analysis to understand the reason of mis-classifications.
- They provide a relatively straight forward method to measure confidence of the models in their predictions.

**Weaknesses:**

Confidence metric:
- A temperature of 0 is used during the confidence estimation. This is practical if you want to evaluate the model on a deterministic response. However, in a practical setting, we do not use low/zero temperature as the models would constraint models diversity. It would be have been more intuitive to see the variation in the metric across different temperatures or at least on the temperature setting shows some level of randomness.
- The number of iteration used is 3, which is probably sufficient given that at temperature 0, the variance in the generated output will be very less. But small variations in the seed values might influence the model predictions. So, I am not entirely sure if the number of iterations would be sufficient for statistically robust confidence.

Clarity in writing:
- While the paper explores interesting research questions like confidence estimation of LLMs, lack of clarity in writing style seriously affects the readability of the paper. I would suggest not using passive voice in sentences and rewriting the content for better clarity should be done before acceptance.

**Questions:**

- Line 201 says "**four** incidents of 2022 were chosen.. " and line 206-207 says  We attributed the **three** different trigger events to two trigger types", was this a mistake ? Or are the **incidents** different from **events**?
- In table 2, there are 2 tables, one showing "Political trigger event" and the other "Sexist trigger event". I am assuming that the total size of the dataset is the sum of the two and you reported the model performance on the combined version of the dataset. Is my assumption right? In any case, this explanation could benefit from a bit more clarity in writing.

---

> ### Author Response · Authors · 2024-11-18
>
> We thank the reviewer for their time and effort in reviewing our work. Their insights were greatly appreciated and provided us with a clearer direction for improvement. While we consider their some of their comments and suggestions in our revised version, we thought it might be best to clarify a few points which may be of interest.
>
> > A temperature of 0 is used during the confidence estimation. This is practical if you want to evaluate the model on a deterministic response. However, in a practical setting, we do not use low/zero temperature as the models would constraint models diversity. It would be have been more intuitive to see the variation in the metric across different temperatures or at least on the temperature setting shows some level of randomness.
>
> We agree with the reviewer’s point on introducing temperature scaling to promote randomness and test the confidence. However, we found that some of the models (such as Alpaca 7B) were susceptible to the temperature scaling and (intuitively so) demonstrated lesser confidence in their prediction, along with varied outputs which varied greatly from the classification categories. To promote simplicity in the use of the confidence measure, we thought of setting the temperature score to a lower value and applying the best model for testing the algorithm. The idea was to ensure that researchers can easily implement this measure to test the reliability of the model’s prediction. Given that LLMs are text generation models, other studies such as Peeperkorn et al. (2024) too state that temperature controls the uncertainty or randomness in the generation process,  we would recommend to keep the temperature to 0 or at a lower value since we want the model to be confident in their prediction if they are used as an annotation system. We do intend to apply randomness in further research with semantic uncertainty measures (Kuhn et al., 2023) which are generally used for natural language generation, such that we can cluster semantically similar predictions in case of varied outputs. But that is beyond the scope of this research.
>
> > The number of iteration used is 3, which is probably sufficient given that at temperature 0, the variance in the generated output will be very less. But small variations in the seed values might influence the model predictions. So, I am not entirely sure if the number of iterations would be sufficient for statistically robust confidence.
>
> We conducted larger variance between the seed values (like 123, 1234, 12345) but to the best of our knowledge, we believe the variance of seed values should not matter as long as they are different each time. In line 413, we mention that three iterations should be fine if all the categories match in the three iterations, otherwise five or more iterations are advised.
>
> The other suggestions are duly noted and would be rectified in the revised version. We thank the reviewer again for their constructive feedback.

---

> > ### Comment · Reviewer_ETQd · 2024-11-26
> >
> > Thanks to the authors for their response. I believe that the paper has its merits, but could benefit from some amount of rework based on all of the review comments. So I would like to stick to my score.

---

### Official Review · Reviewer_tt8A · 2024-11-04

**Soundness:** 3
**Presentation:** 3
**Contribution:** 3
**Rating:** 6
**Confidence:** 4

**Summary:**

This paper presents an evaluation of various LLMs in detecting sexism within political discourse, proposing a new algorithm to measure prediction confidence. The authors discuss the limitations of current LLMs in detecting subtle, context-specific sexism through prompt engineering, suggesting that prompt specificity may not enhance model accuracy (as anticipated).

**Strengths:**

-	This paper is centered around a very interesting topic which is highly relevant given the current online landscape.
-	The corpus will be a useful resource for further research on the subject.
-	Clearly scoped tasks and objectives.

**Weaknesses:**

The paper is generally well written, however there are a few points that must be addressed:

- A section describing existing corpora (annotated for sexism) is missing. The authors should include a brief overview of the existing datasets, including their annotation scheme.

- I believe that since the company was bought, what had previously been free access to Twitter’s API, has been removed. Was the data collected before the cutoff date (~March 2023)?

- At the beginning of Section 3.1.3 the authors state that *’the data was annotated by a group of experts who work on gender studies in political science’*. Until the limitations section, the reader is left to wonder about the IAA and how the final labels were assigned -- these aspects should be mentioned earlier (i.e., in the same section). It would have been good to have even a small sample of the dataset labelled by all the annotators. Does the error analysis reveal any correlations between ‘misclassified’ instances and the person who labelled them?

- On line 201, the authors mention four trigger events, but only three are presented in Table 1.  Considering that on line 907, three trigger points are mentioned, I assume this is the correct number?

- Llama 3 was used for the emotion extraction task – how comes the authors did not include this model too in their analysis?


Suggestions:

- line 217: 1273 -> 1,273
- footnotes should follow (most of the) punctuation marks
- lines 494-496: the ending of the sentence might have to be rewritten
- line 859: stemk -> stem

**Questions:**

Please see the above.

---

> ### Author Response · Authors · 2024-11-18
>
> We thank the reviewer for their time and effort in reviewing our work.
>
> > A section describing existing corpora (annotated for sexism) is missing. The authors should include a brief overview of the existing datasets, including their annotation scheme.
>
> Most of the public databases on online political discourse are heavily US-centric, and the political corpus which were collected are either covered over a span of time, or centred around discussions by politicians, or collected based on political keywords (pertaining to topics such as healthcare or general mentions of political parties). And none of them are labeled for the study of sexism. To overcome this gap, we curated a new dataset fulfilling both criteria.
>
> > I believe that since the company was bought, what had previously been free access to Twitter’s API, has been removed. Was the data collected before the cutoff date (~March 2023)?
>
> The data was collected before the access to data was cut off, which was one of the reasons why we study a period before the data restriction.
>
> > At the beginning of Section 3.1.3 the authors state that ’the data was annotated by a group of experts who work on gender studies in political science’. Until the limitations section, the reader is left to wonder about the IAA and how the final labels were assigned -- these aspects should be mentioned earlier (i.e., in the same section). It would have been good to have even a small sample of the dataset labeled by all the annotators. Does the error analysis reveal any correlations between ‘misclassified’ instances and the person who labeled them?
>
> We had collected annotations in two phases, where we had two annotators label 900 of the tweets in the pilot phase, with one annotator per tweet (this is for the majority of the tweets). In the next phase, we experimented with the remaining number of tweets, having three annotator per tweet, among seven annotators (including the two annotators from before). We had provided them with the annotation guidelines (to be published in the Github repository) and conducted meetings to discuss their task and the purpose of this work. We did not have a large pool of annotators, but the annotators in our study are experts in the field and we believe their annotation offers their perspective. On data curation, we did not see much of a disagreement among the annotators, but we still went for minority voting scheme to promote subjectivity.  In our revised version, we plan to include more information on the annotators in the main text for the benefit of the readers. While prior studies have usually performed majority voting with multiple annotators per instance to mitigate voting bias, there are studies (such as  Fleisig et al. 2024; Basile et al. 2021) which have also explored why majority voting may not be ideal in subjective tasks [like sexism].
>
> > On line 201, the authors mention four trigger events, but only three are presented in Table 1. Considering that on line 907, three trigger points are mentioned, I assume this is the correct number?
>
> There are indeed four trigger points.
>
> > Llama 3 was used for the emotion extraction task – how comes the authors did not include this model too in their analysis?
>
> During the experimentation stage of this work, we did not have access to Llama 3. Since we had already performed the experiments on the other models, we decided to go forward with the analysis of those models. Furthermore, we were not able to complete the prompt stability checks and structure development with new model Llama 3 before completion of the paper. We intend to use the model in our next project. We did see that del Arco et al. (2024) demonstrated impressive results with Llama 3 in calibrating emotions (this information is to be added in the revised version), which we decided to implement to add value to why the political discourse has emotional conflict, given its straight forward implementation.
>
> We thank the reviewer again for their valuable feedback and suggestions. We intend to implement the corrections in the revised version.

---

### Official Review · Reviewer_i5fu · 2024-11-04

**Soundness:** 3
**Presentation:** 3
**Contribution:** 2
**Rating:** 5
**Confidence:** 4

**Summary:**

The work  investigates performance of multiple LLMs in the task of detecting sexism in political discourse and proposes a new algorithm for capturing the confidence of the LLM predictions in classification tasks.

**Strengths:**

-	The paper targets an important task.
-	The paper is clearly written.
-	Extensive experiments were conducted.

**Weaknesses:**

-	There are plenty of papers on detecting misogyny, sexism, etc using LLMs in text. I don’t see political discourse offering specific challenges that requires examining this task in this type of content specifically. Further motivation to this aspect of the work is needed.
-	Line 105: I don’t really see prompt engineering as the reason behind generating good annotations with LLMs, prompt engineering is a tool used to create prompts that can be used to interact with LLMs. The LLMs themselves are the reason/major player in the eventual performance observed in classification for a given task. This confusion continues with line 113, as majority of existing studies focused on investigating the performance of LLMs themselves and not the prompts for NLP tasks.
-	The dataset has some issues that might effect the outcomes of the work.
	-	Data annotation process is somewhat unclear. For example, starting line 174, the two labels/concepts mentioned are described as if they are mutually exclusive, while the work was introduced as detecting sexist language in political discourse. I assume there are many cases where the text is political and sexist. Other missing details are the number of annotators, their gender (as I believe in such task, gender of annotator can bring its own biases to the process), how many annotations were collected per tweet?, etc
	-	The data is really small and targets very few politicians under very specific and small number of trigger events, this of course significantly reduces the studies generalizability for the problem.
	-	Nothing about the quality of the annotations is mentioned. This is key to establish trust in the study built on top of such data.

**Questions:**

-	Line 163: how was this keywords list created? I suggest also sharing/releasing it with the paper.
-	The paper builds a new dataset for the task/up to my knowledge, there are several existing datasets, why not use them?
-	Due to how unbalanced the data is, I suggest not reporting Accuracy. This will also help save space and increase table 3’s font size. Also, prompt category names should be the same as those listed under 3.2.1.
-	In Sec 4, I suggest the core observations/outcomes of each experiment to be better highlighted int ext.

---

> ### Author Response · Authors · 2024-11-18
>
> (Part 1)
>
> We thank the reviewer for taking the time to go through our paper and provide constructive feedback. Here are some points from our side that we think will help the reviewer understand our paper better.
>
> >There are plenty of papers on detecting misogyny, sexism, etc using LLMs in text. I don’t see political discourse offering specific challenges that requires examining this task in this type of content specifically. Further motivation to this aspect of the work is needed.
>
> > The paper builds a new dataset for the task/up to my knowledge, there are several existing datasets, why not use them?
>
> While there has been many papers on detecting sexism and misogyny, to the best of our knowledge, we are yet to have a paper explicitly looking into the sexism in online political discourse. Ziems et al. (2024), Burnham (2024), inter alia has stated political ideology (like quantifying real and perceived political differences) as one of the complex social phenomenon. Political statements directed at women politicians are mistaken as sexist because the emotion/tone in political discourse tends to be negative (given the tone and phrasing of the text, which we show in the appendix) and if negative emotion is directed at women, it is mistakenly classified as sexist. Thus, in the tweets we need to understand which are just political statements directed at women, and then which of them are sexist statements. This is because theoretically and linguistically, political discourse has emotional conflict (which we explore in the appendix) to which it is often difficult to differentiate between the political differences and sexism.
>
> Furthermore, most of the public databases on online political discourse are heavily US-centric, and the political corpus which were collected are either covered over a span of time, or centred around discussions by politicians, or collected based on political keywords (pertaining to topics such as healthcare or general mentions of political parties). And none of them are labelled for the study of sexism. To overcome this gap, we curated a new dataset fulfilling both criteria.
>
> > Line 105: I don’t really see prompt engineering as the reason behind generating good annotations with LLMs, prompt engineering is a tool used to create prompts that can be used to interact with LLMs. The LLMs themselves are the reason/major player in the eventual performance observed in classification for a given task. This confusion continues with line 113, as majority of existing studies focused on investigating the performance of LLMs themselves and not the prompts for NLP tasks.
>
> In general, LLMs have shown to exceed human performance of reliably classifying texts in some domains, without the need for supervision (Ziems et al. 2024; Burnham 2024), which have made them an efficient candidate as an annotation tool. This has led to its widespread adoption among political and social science researchers with its use through prompt engineering using zero-shot and few-shot learning (Burnham 2024a).
>
> In the data annotation process, we intended to provide the definitions we use in our work for the benefit of the annotators, and in no way we aim to indicate the terms to be exclusive of each other. As we stated in the first section of our introduction and again in line 93, it is difficult to identify the presence of sexism in political context, so they are by no means exclusive.

---

> > ### Author Response · Authors · 2024-11-18
> >
> > (Part 2)
> >
> > > The dataset has some issues that might effect the outcomes of the work.
> >
> > We acknowledge that the dataset is small and the number of annotators per tweet is not enough for building annotator trust, but sexism itself is subjective, more so in moderated platforms like Twitter where the form of sexism gets more subtle. Therefore, our aim is not to obtain generalizability but to explore how LLMs perform as annotators in such task in political discourse. Like we mention in the conclusion, through this study we aim to provide informed decisions to researchers who would be using it. In our revised version, we plan to include more information on the annotators in the main text (currently the information is supplied partially in main text and more in the appendix) for the benefit of the readers. While studies have usually performed majority voting with multiple annotators per instance to mitigate voting bias, there are studies (such as  Fleisig et al. 2024; Basile et al. 2021) which have also explored why majority voting may not be ideal in subjective tasks [like sexism]. We did not have a large pool of annotators, but the annotators in our study are experts in the field and we believe their annotation offers their perspective. We had provided them with the annotation guidelines (to be published in the Github) and conducted meetings to discuss the task itself. We use this dataset to test the models’ capability in detecting sexism and we believe the dataset offers some interesting insights which would lead to further studies on the same.
> >
> > > Line 163: how was this keywords list created? I suggest also sharing/releasing it with the paper.
> >
> > The keywords used are the names of the three politicians and their Twitter usernames. We would explicitly mention that in the revised version, and also share the collection strategy in the repository.
> >
> > We thank the reviewer again for the detailed analysis. We agree with some of the points of the reviewer and intend to implement the corrections in the revised version.

---

### Author Response · Authors · 2024-11-26
**Summary of the revised version**

We thank all the reviewers once again for their constructive feedback. We have incorporated most of their suggestions in the revised version. We list a few of the major changes for the benefit of the readers:

- We merged most of the points from the 'Background' section into the 'Introduction' as it contributed to building the narrative for the readers.
- Revised the section on data annotation, and added further information in the appendix.
- We replaced the previous statistical test (logistic regression) for the confidence estimation with another test (point biserial correlation) as it
 fit with the purpose of our hypothesis test better.

Overall, we made a few changes in the document to add more information wherever necessary and delete redundant portions. We also changed the title of the paper, as the current one felt more appropriate for our work.

---

### Meta-Review · Area_Chair_Cm2D · 2024-12-20

**Metareview:**

**Summary:**

The authors evaluate the ability of LLMs to identify sexism in online political discourse. They identified online content referring to 3 female UK MPs and prompted several LLMs to perform either few-shot or zero-shot classification using 4 different prompt designs. They find LLMs struggle to varying degrees with this task. However, confidence estimation experiments indicate that LLMs are very confident about these labeling decisions and have consistently lower confidence for incorrect decisions.

**Strengths:**

- Expert manual annotation of examples for sexism which could be useful to the research community, though there is still ambiguity around the quality of these annotations.

- Reasonably comprehensive comparison across 5 closed and open LLMs.

- Their experiments with uncertainty quantification provide interesting, solid justification for confidence-based measures to identify model errors in sexism detection.

**Weaknesses:**

- The authors provide mostly thorough background/motivation into the challenges to automatic detection of sexist language, though there have been a number of recent works that are not cited here (e.g. [1, 2]) pointing out potential harms in LLM-as-judge settings.

- Narrow scope focused on only 3 women involved in UK politics during 2022.

**Comment:**

In Table 1, the "sexist" example seems to be a reference to other people's sexism (an article sexualizing the politician) rather than the direct speaker being sexist, like the example given at the beginning of the paper. Is any distinction made in this work, because it seems like the two cases should be handled differently?

[1] https://aclanthology.org/2024.emnlp-main.474/

[2] https://arxiv.org/abs/2404.13076

**Additional Comments On Reviewer Discussion:**

The reviews are mixed, and none strongly support either acceptance or rejection. A major concern raised was the scope of the work narrowly focusing on political discourse, and I agree this does seem more appropriate to a niche audience (perhaps an *ACL workshop) rather than ICLR. There were also concerns around determinism in the confidence estimation, which may hinder generalization of findings.

---

### Decision · Program_Chairs · 2025-01-22

Reject